# Zero-Shot Image Enhancement with Renovated Laplacian Pyramid

**Abstract.** In this research, we tackle image enhancement task both in the traditional and Zero-Shot learning scheme with renovated Laplacian pyramid. Recent image enhancement fields experience power of Zero-Shot learning, estimating output from information of an input image itself without additional ground truth data, aiming for avoiding collection of training dataset and domain shift. As requiring "zero" training data, introducing effective visual prior is particularly important in Zero-Shot image enhancement. Previous studies mainly focus on designing task specific loss function to capture its internal physical process. On the other, though incorporating signal processing methods into enhancement model is efficaciously performed in supervised learning, is less common in Zero-Shot learning. Aiming for further improvement and adding promising leaps to Zero-Shot learning, this research proposes to incorporate Laplacian pyramid to network process. First, Multiscale Laplacian Enhancement (MLE) is formulated, simply enhancing an input image in the hierarchical Laplacian pyramid representation, resulting in detail enhancement, image sharpening, and contrast improvement depending on its hyper parameters. By combining MLE and introducing visual prior specific to underwater images, Zero-Shot underwater image enhancement model with only seven convolutional layers is proposed. Without prior training and any training data, proposed model attains comparative performance compared with previous state-of-the-art models.

**Keywords:** Zero-Shot Learning, Underwater Image Enhancement, Laplacian Pyramid, Image Restoration

## 1 Introduction

Image enhancement is essential in measuring physical environment and progressively has been improved with deep learning. Previous deep learning models mainly focus on supervised approaches by mapping degraded images to clear images employing large scale image dataset in various image enhancement fields [25, 26, 39]. However, constructing real large scale image pairs requires tremendous costs, as well as obtaining massive clear images is inherently difficult [25] or sometimes impossible in fields like underwater image processing [27]. Alternatively employed artificial dataset based on physical model or generative adversarial network (GAN) [19, 28, 40] suffers from domain shift, as artificial images are less informative and may be apart from real images, resulting in limited capability of deep learning compared with other image processing tasks [3, 25].

**Fig. 1.** Example outputs of proposed MLE. Just enhancing an image in Laplacian pyramid representation, contrast improvement (left), detail enhancement (middle), and underwater image enhancement (right) is performed in the same MLE scheme.

As requiring "zero" training data, Zero-Shot image enhancement has been getting a lot of attention and promising results are shown in denoising [24], super-resolution [11], dehazing [25], back-lit image restoration [44], and underwater image enhancement [22]. Note that Zero-Shot image enhancement recovers images only from information of an input image itself, different from typical term in general classification task [25]. As no ground truth is available, constructing task specific loss function is especially important in Zero-Shot image enhancement. Currently, loss function is mainly introduced to reflect effective prior or bias of underlying phenomena or some specific task, followed by getting feedback from parameterized latent physical model or knowledge of the target task [14, 25, 38]. To be specific, based on the widely used atmospheric scatter model [30, 31], efficient and clear output is obtained in dehazing task [25] with loss function reflecting internal physical process composed of global atmospheric light, transmission map, and statistical property, Dark Channel Prior [18]. In back-lit image restoration, luminance is adjusted with loss of parameterized s-curve function after being mapped to YIQ color space [44]. In underwater image enhancement, Zero-Shot learning based model on the Koschmieder's physical model is first proposed in [22]. Loss function for modifying the inherent property of an image like smoothness or color balance is also effective [8, 38].

On the other, though supervised deep learning models have improved by incorporating traditional signal processing methods such as discrete wavelet transform, whitening and coloring transform, and Laplacian pyramid, less exists in Zero-Shot learning. In order to further accelerate Zero-Shot image enhancement, this research proposes to incorporate traditional Laplacian pyramid [7] to network process. First, Multiscale Laplacian Enhancement (denoted as MLE) is formulated, which simply convolves and enhances images in multiscale Laplacian pyramid representation, depending on three hyper parameters, kernel size $K$ and standard deviation $\sigma$ of Gaussian kernel for constructing Laplacian pyramid, and pyramid level $L$. Compared to simple convolution, enlarged receptive field which operates to an image is naturally formulated in MLE. By employing unsharp masking filter in MLE scheme, impacts of detail enhancement, image sharpening, and contrast improvement are experimentally shown (Figure 1), de-

pending on its hyper parameters. Though the effectiveness of MLE is observed, hyper parameter tuning of MLE is practically inconvenience. Accordingly, Zero-Shot Attention Network with Multiscale Laplacian Enhancement (denoted as ZA-MLE) is proposed to integrate several enhanced results of MLE. ZA-MLE consists of only seven convolutional layers and process for enhancing multiscale features of an image, implemented with element wise product and addition of a contrast improved result of MLE and a sharpened result of MLE (denoted as Zero-Shot Attention). Also, we introduce "prior" specific to underwater image enhancement for selecting MLE. Namely, we experimentally found that the top component of Laplacian pyramid of a degraded underwater image (2nd column of Figure 2) contains less original signal information, thus just removing the top component tends to extract original, high frequency signal (3rd column of Figure 2). Despite requiring no training data and prior training, proposed ZA-MLE mainly achieves comparative performance compared to other latest supervised models in underwater image enhancement. Also compared to previous Zero-Shot learning based model [22], our ZA-MLE advantageously works fast thanks to its simple structure, as well as quantitative scores improve. Our main contributions are summarized as follows: 1. Formulation of MLE simply convolving and enhancing an image in multiscale Laplacian pyramid representation. Depending on hyper parameters, effects of detail enhancement, image sharpening, and contrast improvement are shown. 2. Propose of ZA-MLE. To the best of our knowledge, ZA-MLE is first proposed Zero-Shot learning based underwater image enhancement model combining traditional signal processing method, Laplacian pyramid and unsharp masking filter. Though working in Zero-Shot manner, proposed ZA-MLE achieves favorable performance compared to latest models. 3. Propose of elaborated loss function for Zero-Shot learning. Gradient domain loss as well as color correction loss and reconstruction loss are combined.

## 2    Related Work

### 2.1    Traditional Signal Processing Method and Deep Learning

In this section, we state the relationship between conventional signal processing methods and deep learning. Rapid advancement of deep learning is accelerated by construction of large scale dataset and sophisticated very deep network architecture [17]. More recently, deep learning architecture has further developed by incorporating traditional signal processing methods. To be specific, discrete wavelet transform is employed to preserve fine structure of an input, by passing high frequency components extracted in the encoder part to the decoder part [13,43]. In underwater image enhancement, white balance, histogram equalization, and gamma correction are combined to mitigate domain shift between training data and test data [27]. In style transfer, content features are projected to style features with whitening and coloring transform [12]. To the best of our knowledge, these hybrid deep learning models are basically limited to supervised learning and integrating Zero-Shot underwater image enhancement especially with Laplacian pyramid is first proposed in this research.

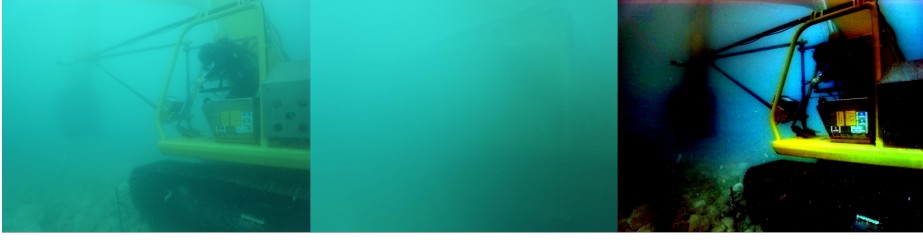

**Fig. 2.** Input image (1st column), the top Laplacian pyramid component of the image (2nd column), and result of subtraction of the top component from the original input (3rd column). We experimentally found that the top Laplacian pyramid component of a degraded underwater image dominates noise signal, and just removing the component tends to extract original signal. The 3rd column is multiplied by ten for visualization.

### 2.2 Laplacian Pyramid and Image Restoration

Multiscale feature of an image is valuable and efficiently extracted with Laplacian pyramid [4, 7]. Laplacian pyramid hierarchically presents an image as a sum of band-pass images depending on resolution or frequency, practically obtained by subtracting adjacent elements of Gaussian pyramid constructed from the original image [7, 33]. An image of high frequency features as edge or contour mainly present in low pyramid levels, while low frequency features like color is decomposed in high pyramid levels [33]. Laplacian pyramid is employed in loss function [5, 41] or is combined with network architecture [20] to reflect various image features in supervised learning.

## 3 Multiscale Laplacian Enhancement for Image Manipulation

### 3.1 Formulation of Multiscale Laplacian Enhancement

We formulate Multiscale Laplacian Enhancement, denoted as MLE. MLE simply, yet efficiently enhances multiscale features of an image in Laplacian pyramid domain, owing to its hierarchical image representation reflecting frequency or resolution. The detailed derivation is found in the supplementary material.

Let an input image be $I(x)$, where $x$ is the position of the image. An input image is first divided into Laplacian pyramid representation and each pyramid elements are simply filtered followed by the reconstruction phase, formulated as:

$$MLE\left[I(x)\right] = \sum_{i=0}^{N-1}\left\{U^i\left[L_i\left[I(x)\right]*Filter\right]\right\} + U^N\left[R_N*Filter\right]$$

where pyramid level $N$, a convolution $Filter$, and up-sampling operator $U$ should be set beforehand. Here, $L_i(x)$ means the i-th level of Laplacian component obtained by subtracting adjacent elements of Gaussian pyramid constructed from the original image [7, 33], defined as $L_i\left[I(x)\right] := \left(D \circ G\right)^i\left[I(x)\right] -$

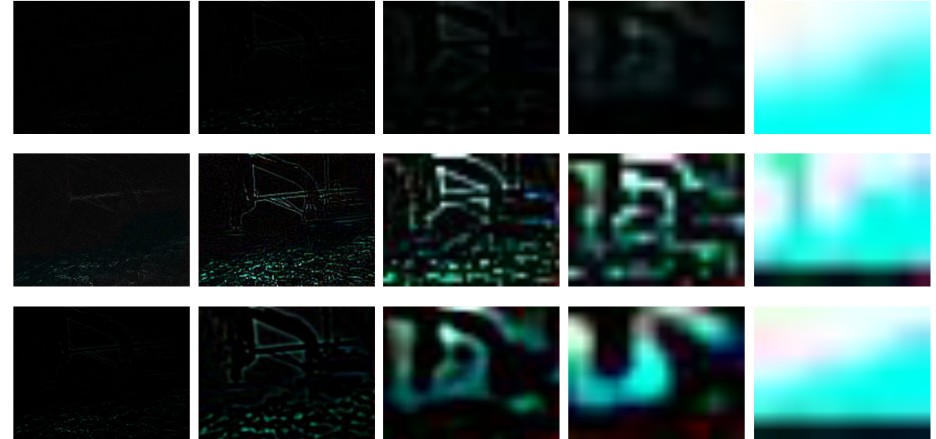

**Fig. 3.** Elements of Laplacian pyramid constructed from an original image (1st row), MLE-733 (2nd row), and MLE-799 (3rd row). From left to right, results of pyramid level $N = 1, 3, 5, 6, 7$ are shown. MLE-733 more enhances high frequency signal like edge or texture in lower pyramid levels, while MLE-799 more enhances low frequency signal like color in higher pyramid levels. Results are multiplied by ten for visualization.

$G \circ (D \circ G)^i [I(x)]$. Above $G$ and $D$ mean low-pass Gaussian filter and downsampling operator by a factor of 2, respectively. $R_N$ is the top component (lowest resolution) of the Laplacian pyramid. Each Laplacian components are upsampled to the original resolution of the input after filtering.

Compared with normal convolution, image size to which filters operate is different from MLE. To be specific, while MLE filters various resolution of an image of each Laplacian pyramid components, normal convolution operates only in the same size of the input, expressed as follows:

$$I(x) * Filter = \sum_{i=0}^{N-1} \left\{ U^i \left[ L_i \left[ I(x) \right] \right] * Filter \right\} + U^N \left[ R_N \right] * Filter$$

The above equation is directly obtained using linearity of convolution and Laplacian pyramid representation of the input. Enlarged effective filter size is practically useful in extracting image features [10], which is naturally formulated in MLE. With the simple idea and implementation of MLE, multiscale features of an input image is efficiently enhanced.

### 3.2   Internal Results of Multiscale Laplacian Enhancement

MLE is simply defined to filter an image in Laplacian pyramid domain, resulting in efficient enhancement of overall multiscale features of an image with naturally formulated enlarged receptive field. Here, MLE depends on three hyper parameters, pyramid level $N$, kernel size $K$, standard deviation $\sigma$ of Gaussian kernel,

**Table 1.** Sharpness [15] and RMS contrast [34] of PIPAL [21] and DIV2K [1] dataset.

| Dataset | Metric | MLE-133 | MLE-333 | MLE-599 | MLE-799 | UNSHARP | RAW |
|---------|--------|---------|---------|---------|---------|---------|-----|
| PIPAL [21] | RMS contrast | 0.102 | 0.113 | 0.108 | 0.112 | 0.094 | 0.075 |
| | Sharpness | 2.961 | 2.799 | 2.692 | 2.615 | 2.768 | 1.063 |
| DIV2K [1] | RMS contrast | 0.085 | 0.094 | 0.091 | 0.098 | 0.079 | 0.069 |
| | Sharpness | 2.171 | 2.084 | 1.963 | 1.914 | 1.989 | 0.723 |

denoted as MLE-$NK\sigma$ in order. In order to examine the effectiveness of MLE, we use basic unsharp masking filter for MLE, traditionally employed in image sharpening task [36], denoted as: $Filter_{unsharp} := \left( \begin{smallmatrix} -1 & -1 & -1 \\ -1 & 9 & -1 \\ -1 & -1 & -1 \end{smallmatrix} \right)$. Selecting other sophisticated filters is our future work. Throughout numerical experiment, bicubic up-sampling is employed.

First, to comprehend the behavior of MLE scheme, the enhanced results of each Laplacian pyramid levels of an image are shown in Figure 3. The first row shows elements of an original image, while the second and the third row show results of MLE-733 and MLE-799, respectively. Results of $N = 1, 3, 5, 6, 7$ are respectively shown from left to right. Each figures are multiplied by ten for visualization. Compared with the first row, various image features, contour captured in low pyramid levels and slightly appeared color signal in high pyramid levels, are efficiently enhanced. Also comparing the 2nd and 3rd row, each elements employing $K = 3$, $\sigma = 3$ (2nd row), preserve high frequency signal thus clearly sharpened, while results of $K = 9$, $\sigma = 9$ (3rd row), relatively cut off high frequency signal, emphasize color information more (4th column). Laplacian components of an original image (1st row) is dark and hardly be seen.

## 3.3   Comparison with Unsharp Masking Filter

Next, we proceed to comparison with conventional unsharp masking filter generally employed in image sharpening [36]. Qualitative and quantitative results of unsharp masking filter are respectively shown in the 4th column of Figure 4 and Table 1. We utilize reference images from PIPAL dataset [21] and high resolution images from DIV2K dataset [1], mainly employed in image restoration task. 2nd and 3rd column of Figure 4 respectively present results of MLE-133 and MLE-799, while 1st column presents original input images, denoted as RAW. Detail enhancement or edge emphasis is performed with MLE-133, while sharpness slightly improved with unsharp masking filter (1st row of Figure 4). Note that filtering is performed totally twice in MLE-133. Compared with MLE-799, 3rd column, characteristically vivid, contrast enhanced results are obtained. As image features of higher pyramid levels are also enhanced, the overall sharpening effect of MLE-799 is weaker than MLE-133. For quantitative evaluation, Sharpness [15] and RMS contrast [34] are evaluated. Sharpness is the strength of vertical and horizontal gradient after Sobel filtering, and RMS contrast means the standard deviation of luminance intensities. In quantitative results from Table 1, all metrics are improved with MLE and unsharp masking filter from the original image. Sharpness is the highest in MLE-133 and decreased with higher

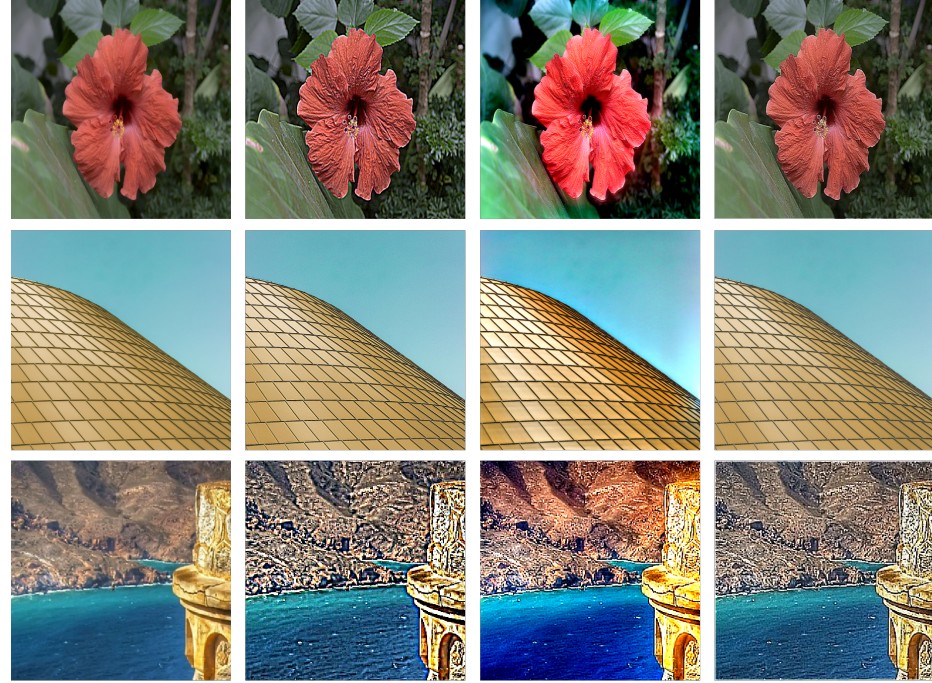

**Fig. 4.** Comparison with MLE and unsharp masking filter. 1st column shows input images, 2nd and 3rd column shows results of MLE-133 and MLE-799, respectively. 4th column shows results of unsharp masking filter. Image sharpening or detail enhancement is performed in MLE-133, while contrast improvement is performed in MLE-799.

pyramid level, as Sharpness measures image gradient and relatively higher in MLE-133 emphasizing only high frequency features. MLE-333 gets the highest score for RMS contrast in PIPAL dataset [21], while MLE-799 is the 1st in DIV2K dataset [1]. As RMS contrast measures dispersion of luminance of an image, MLE-799, also enhancing higher pyramid components, got higher scores. Image sharpening, detail enhancement, and contrast improvement are performed in the same MLE scheme depending on its hyper parameters.

## 3.4   Ablation Study of MLE

In this section, results of different parameter settings of MLE, namely, pyramid level $N$, kernel size $K$, and standard deviation $\sigma$ of Gaussian kernel for constructing Laplacian pyramid are evaluated. Results of different pyramid levels are shown in Figure 5. Input image (4th column) is enhanced with MLE-133 (1st column), MLE-333 (2nd column), and MLE-733 (3rd column). As we confirmed in the 2nd and 3rd columns of Figure 4 and Table 1, high frequency signal like configuration or edge of banked up rock is sharpened with MLE-333, while contrast is improved with MLE-733. The number of filtering as well as resolu-

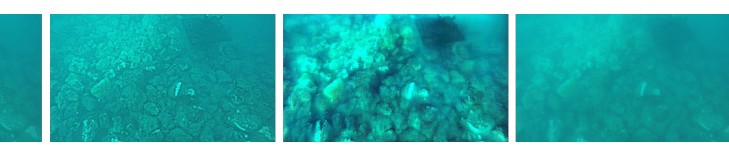

**Fig. 5.** Results of different pyramid levels of MLE. From left to right, images of MLE-133, MLE-333, MLE-733, and an input image are shown, respectively.

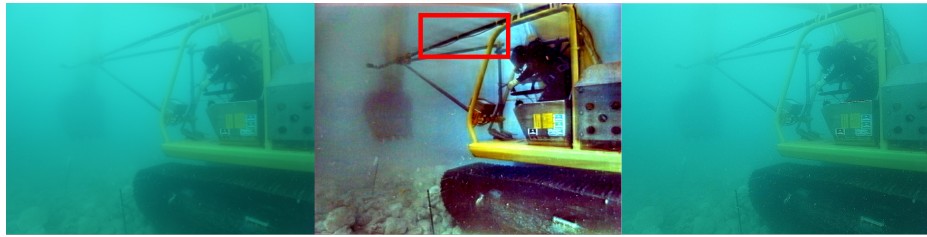

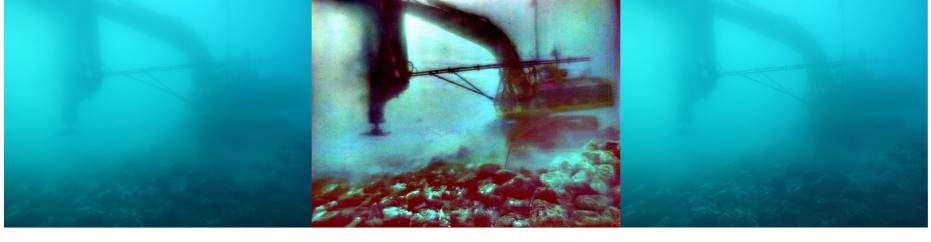

**Fig. 6.** Comparison with unsharp masking filter (3rd column). Input blurred underwater images (1st column) are enhanced with MLE-833 (2nd column).

tion of pyramid components to which filters operate changes in accordance with pyramid levels, resulting in enhancement of various image features of an image.

As for $K$ and $\sigma$ of Gaussian kernel, the lower $K$ and $\sigma$ are, the smaller the effect of blurriness of filtering, as a consequence, wide range of frequency band tends to preserve also in higher pyramid levels, thus is strongly enhanced. Qualitative results are found in the supplementary material. Also refer to the 2nd and 3rd rows of Figure 3. The degree of enhancement basically depends on contrast or sharpness of an original input as well as hyper parameters of MLE. As for clear land images, we experimentally observe that $K = 9$, $\sigma = 9$ are usually optimal setting for contrast improvement, and $K = 3$, $\sigma = 3$, $N = 1$ for image sharpening or detail enhancement. Practically, hyper parameters of MLE needs to be selected depending on objective task or input sharpness.

### 3.5    Application of MLE to Underwater Images

We proceed to application of MLE to underwater images. As many underwater images suffer from lowering of contrast or blurriness, MLE favorably sharpens severely degraded underwater images. Compared to results of unsharp masking

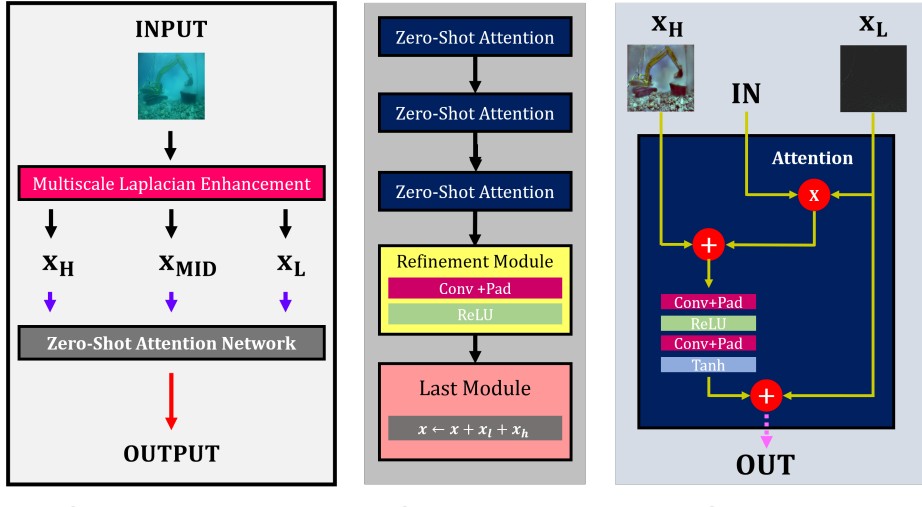

**Fig. 7.** Overall processing flow of ZA-MLE (left), Zero-Shot Attention Network (middle), and Zero-Shot Attention Module (right). 1: An input is first processed with different parameter settings of MLE, resulting in $X_H$ from MLE-753 and $X_L$ from MLE-331. Note that the top pyramid component of $X_L$ is removed to enhance high frequency signal. $X_{MID}$ is calculated with $X_{MID} := X_H * X_L + X_L$. 2: Input $X_L$, $X_H$, and $X_{MID}$ into Zero-Shot Attention Network, aiming for enhancing multiscale feature of an input in Zero-Shot manner. 3: Output is obtained after training of 300 epochs. Different from elaborated learning strategy [39], we stop training in 300 epochs and fixed.

filter shown in Figure 6, severely degraded underwater images (1st column) are prominently recovered with MLE-833 (2nd column), though traditional unsharp masking not (3rd column). Qualitative behavior of MLE is similar both in underwater and land images, though underwater images are usually low contrast and blurred. While effectiveness of MLE is also observed in underwater images, halo sometimes appears in some MLE results like 2nd row of Figure 4 or 1st row of Figure 6, depending on image sharpeness and hyper parameters of MLE, caused by linearity of filter [33]. In this research, we set widely utilized linear unsharp masking filter to confirm effectiveness of MLE, and incorporating edge preserving, more sophisticated filter to MLE is our future work.

## 4    Zero-Shot Attention Network with Multiscale Laplacian Enhancement (ZA-MLE)

As we discuss in previous sections, results of MLE depend on input image sharpness or hyper parameters, which is practically troublesome. For the convenience in practical application, we propose simple, yet efficient Zero-Shot image en-

hancement scheme combined with MLE and elaborated loss function for underwater image enhancement without training data and prior training.

## 4.1   Process of ZA-MLE

Overall processing flow of Zero-Shot Attention Network with Multiscale Laplacian Enhancement (denoted as ZA-MLE) is shown in Figure 7. Input image is first processed with different parameter settings of MLE, resulting in $X_H$ and $X_L$, followed by Zero-Shot Attention Network. In MLE part, $X_H$ is designed to improve contrast of an input underwater image and MLE-753 is experimentally selected. As for $X_L$, MLE-331 is experimentally set for extracting high frequency signal of an image like edge or configuration. Note that the top pyramid component, low frequency signal, of MLE-331 is removed in reconstructing $X_L$ to enhance high frequency signal of an original input, inspired by the insight shown in Figure 2. Here, $X_{MID} := X_L * X_H + X_L$ is employed as an input of ZA-MLE. After obtaining $X_H$ and $X_L$, inspired by [37], element wise product and addition follow, which is designed to sharpen a contrast improved $X_H$ with attention of sharpened $X_L$ to integrate enhanced results of MLE, denoted as Zero-Shot Attention. After Zero-Shot Attention, $X_H$ is added followed by layers of one convolution, Leaky ReLU activation, one convolution, and Tanh activation (right in Figure 7). The number of channels are increased from three to six in the first convolution, and decreased from six to three in the second convolution. $X_L$ are finally added to enhance low frequency features. This procedure is denoted as Zero-Shot Attention module. After three Zero-Shot Attention modules, Refinement Module consisting of one convolution and Leaky ReLU activation layers follows. Then, $X_H$ and $X_L$ are finally added to incorporate multiscale features of an image, denoted as Last Module. As each Zero-Shot Attention module contains only two convolutional layers, proposed ZA-MLE totally includes seven convolutional layers. With the power of proposed MLE, simply implemented ZA-MLE enables efficient image enhancement of challenging real underwater images.

## 4.2   Loss Function

The loss function for training ZA-MLE consists of three terms, reconstruction loss $l_{rec}$, derivation loss $l_{deriv}$, and color loss $l_{col}$, defined as follows:

$$Loss = \alpha l_{rec} + \beta l_{deriv} + \lambda l_{col}$$
$$l_{rec} := \|X_{out} - X_H\|_1$$
$$l_{deriv} := \| \left(\partial_x^2 (X_{out}) - \partial_x^2 (X_L)\right) + \left(\partial_y^2 (X_{out}) - \partial_y^2 (X_L)\right) \|_1$$
$$l_{col} := \|R_{out} - G_{out}\|_1 + \|B_{out} - G_{out}\|_1$$

where $\partial_x^2$ and $\partial_y^2$ respectively compute 2nd order horizontal and vertical gradients. $R_{out}$, $G_{out}$, and $B_{out}$ respectively mean R, G, B channels of $X_{out}$.

The reconstruction loss $l_{rec}$ works as regularization term and defined as l1 distance between model output $X_{out}$ and $X_H$, in order to get output similar to $X_H$ to incorporate MLE. Note that reconstruction loss $l_{rec}$ in this research is

**Table 2.** Results of UIQM and UCIQE. Proposed ZA-MLE without training data and prior training achieves comparative results compared with other latest supervised and Zero-Shot learning based models. Scores of ZA-MLE show the average of three trials.

| Dataset | | ZA-MLE | UWCNN [26] | Water-Net [27] | U-Transformer [35] | All-in-One [29] | Koschmieder [22] |
|---------|------|--------|------------|----------------|--------------------|-----------------|------------------|
| Challenging-60 [27] | UIQM | 2.754 | 2.386 | 2.609 | 2.724 | 2.679 | 2.402 |
| | UCIQE | 4.986 | 3.203 | 4.554 | 4.231 | 4.690 | 4.170 |
| Original | UIQM | 2.989 | 2.558 | 3.031 | 2.664 | 2.900 | 1.946 |
| | UCIQE | 5.424 | 4.127 | 6.057 | 5.313 | 2.998 | 5.392 |

different from [38] which employs an original input image $X_{in}$ instead of $X_H$, to get output similar to $X_{in}$. The derivation loss $l_{deriv}$ is l1 distance between $X_{out}$ and $X_L$ in gradient domain [9] designed to reflect gradient information of $X_L$ and inhibit noise. For correction of color distortion of underwater images, based on the gray world assumption, white balance is performed to enforce each RGB channels to have the same values as in land images [6, 38]. Modified from the original gray world assumption, proposed color correction loss $l_{col}$ is implemented to enforce R and B channels to have similar values to G channel of $X_{out}$, as G channel of an underwater image is less susceptible to underwater conditions [2].

   Input image processed with MLE is passed through Zero-Shot Attention Network and the network is trained based on the above loss function employing $X_L$ and $X_H$. Note that $X_L$ and $X_H$ are obtained from an input image itself thus requiring no training data. We experimentally observe that visually pleasing result is obtained around 200 to 300 epochs as in following section.

## 5   Experiment

### 5.1   Experimental Setting

In this section, we evaluate the performance of our ZA-MLE for underwater image enhancement. We first initialized network parameters of ZA-MLE with [16], and used Adam optimizer [23] with the learning rate 0.001. Unlike previous elaborated learning strategy [39], we stop training at 300 epochs, experimentally selected, and fixed throughout the experiment. ZA-MLE works as Zero-Shot manner, trained per an input image without additional data or prior training. For practical application, prior training is recommended for more speed up. As discussed in previous section, hyper parameters of MLE, pyramid level, kernel size, and standard deviation, are respectively set as MLE-753 for $X_H$ and MLE-331 for $X_L$ in order. $X_{MID}$, defined as $X_{MID} = X_L * X_H + X_L$, is employed for the input. As for real underwater image dataset, Challenging-60 (4th to 6th column of Figure 8) [27] and the Original dataset containing notably deteriorated 77 images taken in Okinawa, Japan (1st to 3rd column of Figure 8), are employed for the evaluation. In evaluating recovered results, generally utilized non-reference metric, UIQM [32] and UCIQE [42] designed to reflect human perception, are computed as existing no ground truth for real images. Coefficients of loss functions, $\alpha$, $\beta$, and $\lambda$ are respectively set to 1.0, 1.0, and 0.1. We implement our model with PyTorch and GeForce RTX 2080 Ti GPU.

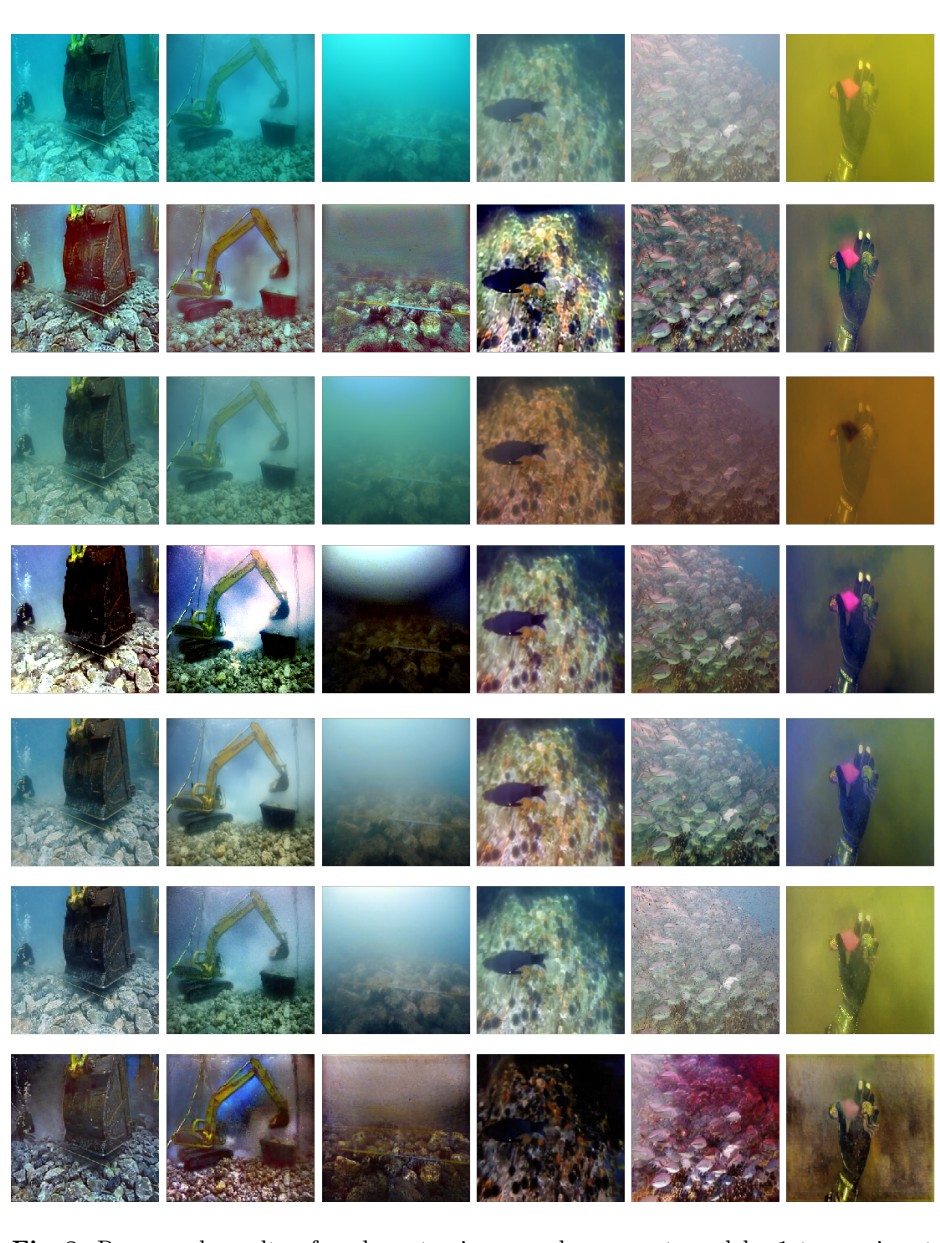

**Fig. 8.** Recovered results of underwater image enhancement models. 1st row: input raw images. 2nd row: results of ZA-MLE. 3rd row: results of UWCNN [26]. 4th row: results of Koschmieder [22]. 5th row: results of Water-Net [27]. 6th row: row: results of U-Transformer [35]. 7th row: results of All-in-One [29].

## 5.2   Results and Discussions of ZA-MLE

We compare our ZA-MLE with state-of-the-art underwater image enhancement methods. Currently, as Zero-Shot learning based model is rare [22], available supervised models are also compared. First row of Figure 8 shows input raw underwater images, and the second row shows results of proposed ZA-MLE. 4th row shows results of Zero-Shot model, denoted as Koschmieder [22]. The rest row show results of supervised models, 3rd row: UWCNN [26], 5th row: Water-Net [27], 6th row: U-Transformer [35], and 7th row: All-in-One [29].

In UIQM [32] and UCIQE [42] scores of Table 2, our ZA-MLE achieves favorable performance, getting first rank in the Challenging-60 dataset [27] and second rank in the Original dataset, compared with other supervised models which require training data or prior training. In quantitative results of Figure 8, ZA-MLE (2nd row) basically corrects blueish (1st and 2nd column) and even yellowish (6th column) color cast, as well as visibility of blurred underwater images (3rd to 5th column) improves. Among previous methods, CNN based Water-Net combining white balance, histogram equalization, and gamma correction [27] (5th row), qualitatively and quantitatively recovers well. Owing to conducting white balance to an input image, this network recovers also a yellowish image (6th column), yet not sufficiently sharpens deteriorated images and somewhat blurred. By contrast, our ZA-MLE sharpens blurred images thanks to proposed MLE and Zero-Shot Attention. Results of U-Transformer [35] (6th row), Vision Transformer based model, are relatively well, but fail to recover yellowish image (6th column) and sometimes adds grid artifacts (5th column). UWCNN [26] and All-in-One [29] hardly improve visibility. Supervised models often suffer from domain shift between training data and test data caused by complex real underwater environment, still a challenging issue [3].

Next, comparison with results of the Koschmieder [22] (4th row of Figure 8) are shown. Currently, to the best of our knowledge, Zero-Shot learning based underwater image enhancement model does not exist except [22]. Code and parameter setting is directly employed provided by the authors, setting epoch size to 10000. In terms of UIQM and UCIQE in Table 2, proposed ZA-MLE performs better than Koschmieder [22] in both dataset. In quantitative comparison, [22] corrects blueish and yellowish underwater images based on the Koschmieder's physical model, yet also outputs a little over enhanced results (1st, 2nd, and 3rd column of Figure 8). Specifically, color distortion is sometimes observed as in the 2nd column. As [22] mainly corrects color cast based on the Koschmieder's model, recovered results of severely degraded underwater images are not very good, which do not likely to obey the Koschmieder's model.

Our ZA-MLE is different from [22] in that incorporating traditional signal processing method, Laplacian pyramid and unsharp masking filter. In terms of calculation time, owing to its simple structure, ZA-MLE, trained with 300 epochs, costs about 2 seconds, while Koschmieder, trained with 10000 epochs, costs about 5 minutes to process $256 \times 256$ images, 150 times faster than Koschmieder. Time analysis is conducted with NVIDIA 2080Ti GPU and Intel Core i9-9900K CPU.

**Table 3.** UIQM and UCIQE scores calculated from different loss functions and dataset.

| Loss | Dataset | Challenging-60 | Original |
|------|---------|----------------|----------|
| $l_{rec}$ | UIQM/UCIQE | 2.749/4.946 | 2.957/5.273 |
| $l_{rec} + l_{deriv}$ | UIQM/UCIQE | 2.770/4.987 | 3.029/5.442 |
| ALL | UIQM/UCIQE | 2.772/4.967 | 3.030/5.394 |

**Table 4.** Comparison of UIQM and UCIQE scores of two color loss functions. Average scores of 10 trials are shown.

| Loss | Dataset | Challenging-60 | Original |
|------|---------|----------------|----------|
| Propose | UIQM/UCIQE | 2.748/4.978 | 2.987/5.361 |
| Gray-world [6] | UIQM/UCIQE | 2.748/4.961 | 2.987/5.350 |

### 5.3   Ablation Study of Loss Function

Ablation study of the loss function combining three terms is shown in Table 3. UIQM and UCIQE scores of the loss function employing $l_{rec}$, $l_{rec} + l_{deriv}$, and all terms are compared. In terms of UIQM, weighting contrast or sharpness more, all terms contribute in all dataset. As for UCIQE, weighting chroma or saturation of an image, while $l_{deriv}$ improves the score, color loss $l_{col}$ a little decreases UCIQE in both datasets. $l_{col}$ is designed to balance color channels of underwater images, making R and B channels close to G channel. Decreased UCIQE especially in the Original dataset might be caused by decreased chroma of an output, as Original dataset more includes greenish underwater images.

Proposed $l_{col}$ is modified from the original gray world assumption [6], which enforces all channels to be the same, and UIQM and UCIQE scores are shown in Table 4. Proposed $l_{col}$ slightly improves the scores of UCIQE. We observe that $l_{col}$ also contributes to learning stability and adopted.

## 6   Conclusion

This research proposes a simple, yet efficient Zero-Shot image enhancement scheme incorporating traditional signal processing method, Laplacian pyramid. First, MLE just convolving and enhancing images in multiscale Laplacian pyramid representation is formulated. Combined with basic unsharp masking filter in MLE scheme, the effects of image sharpening, detail enhancement, and contrast improvement are shown depending on its hyper parameters. Combining MLE to network process, ZA-MLE is also proposed to enhance underwater images trained with the elaborated loss function. To the best of our knowledge, Zero-Shot learning based underwater image enhancement model incorporating Laplacian pyramid is first proposed in this research. By reflecting visual prior specific to underwater images, simply implemented ZA-MLE achieves comparative performance compared to other latest deep learning models.

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
