# OpenReview forum: "Zero-Shot Image Enhancement with Renovated Laplacian Pyramid"
_thecvf.com/ECCV/2022/Workshop/VIPriors — VIPriors 2022 OralPosterTBD_

### Official Review · Reviewer_o9YT · 2022-08-04
**Well-written paper, in general good work, impressive underware image enhancement visualization**

**Rating:** 7
**Confidence:** 4

**Review:**

In general, this paper is a good work for this workshop. It renovates Laplacian pyramid for zero-shot image enhancement. And also propose a zero-shot attention network with multiscale Laplacian enhancement. The qualitive results, e.g., under water images enhancement, are impressive and good. The authors conduct image enhancemnet experiments on mainly two datasets and evaluate the results with two metrics.

Question:
Authors mention that MLE depends on hyperparameters, N, K, $\sigma$. In section 3.4 and supplementary material, only limited qualitative results are provided. Do you have any quantitative results to prove that?

Minor: Line 77, should "on the other" be ‘on the other hand’?

---

### Official Review · Reviewer_to9H · 2022-08-05
**The paper proposes a method that incorporates the traditional Laplacian pyramid method with a Zero-shot approach. The method provides convincing results on image quality in terms of detail, sharpness, colour and contrast.**

**Rating:** 10
**Confidence:** 5

**Review:**

#Summary:

The paper proposes a method that incorporates the traditional Laplacian pyramid method with a Zero-shot approach. The method provides convincing results on image quality in terms of detail, sharpness, colour and contrast.

#Strengths:
- Clarity and rigorousness
- Performance
- Ablation studies
- Experiments on various input domains
- The form of the loss function

#Weakness:
- Applying ZA-MLE only on underwater images. It could be nice to apply it to normal image enhancement tasks and we could see the generalizability of the method.

#Some Questions and Remarks:
- Fig 2: The method enhances the details, for sure, but also brings some artefacts. For instance, I cannot identify the scuba diver in the right image anymore and it looks like there is a dark region. If I run an object detector on that images, probably the detector misses him.
- How does the method perform under very dark and noisy scenes? X_L and X_H are obtained from input images, I was wondering if the zero-shot technique might work under extremely dark and noisy setups.
- What is the reason/motivation to use only X_H for l_rec calculation?
- Will the code be publicly available?


#Suggestions:
- Fig 1: Putting an emphasis on the lines in the middle of images and giving their reasons with a sentence in the caption can help the reader to realise the differences.
- Table 1: Adding colour indicator information (blue, red) in the image caption.
- Fig 6: Mentioning the reason for the red box in the image caption too.

---

### Decision · Program_Chairs · 2022-08-08

**Decision:**

Accept (Oral/Poster TBD)

**Comment:**

Dear authors,


Congratulations! Your work has been accepted to the VIPriors workshop. Decisions on oral/poster presentations will follow later, when the program of the workshop is finalized.

*Please note the first action item is due on Wednesday! Please see instructions below.*

**Camera-ready instructions**

There is some work left to be done to ensure your work is included in the ECCV conference workshop proceedings. The ECCV publication managers use CMT to collect all workshop papers. This means we will migrate your paper from the VIPriors OpenReview page to the centralized ECCV workshop proceedings CMT page. The VIPriors program committee will ensure the details of your work (name, title, email address) are transferred to the CMT page, after which the ECCV proceeding managers will invite you to upload the camera-ready version of your work to the centralized ECCV CMT workshop proceedings page.

Please carefully follow the following instructions:
- **Before August 10th**, ensure that the first author has a CMT account under the same email address as the OpenReview account through which the accepted work was submitted. This account will be used to invite you to upload the camera-ready paper.
- Fill out this form, to inform us that the CMT account is in order: https://docs.google.com/forms/d/e/1FAIpQLSfyAoPv2_srESKaLRHIsHoWe3Fss1Z50ykdH7SzZpenA0m_5g/viewform
- Await instructions from the ECCV publication organizers, sent through CMT, on how to submit your camera-ready paper.
- Submit the camera-ready paper **before August 22nd**. Follow the camera-ready instructions for the main conference: https://eccv2022.ecva.net/submission/call-for-papers/.

**Attending the workshop**

We invite all authors of accepted works to attend the workshop in person on October 24th 2022 at ECCV in Tel Aviv. Please note a conference registration is required to attend the workshop. The workshop will be hybrid, enabling both in-person and remote attendance. We hope all accepted works can be represented in-person by at least one author, but we understand if this is not possible. Remote attendance of the workshop will be possible, though unfortunately there are limits on presenting works remotely: we intend to enable remote oral presentations, but this is not possible for posters.

Please fill out this form *before September 26th* to inform us of your attendance: https://docs.google.com/forms/d/e/1FAIpQLSfqRhdd2pq8t4CC8hL_c8fQo_TWcbzuQH3KGLzKVE36iTW_oQ/viewform.

**Presenting your work at the workshop**

Authors of all accepted papers are invited to present a poster at the workshop. Instructions on poster format will follow at a later date, but we will ask you to print and bring your own poster to the workshop.


For more information, as well as updates on the program of the workshop, keep an eye on our website: https://vipriors.github.io.

We thank you for choosing to submit to our workshop, and we are very much looking forward to hosting you in person in Tel Aviv!


Kind regards,

Robert-Jan Bruintjes
VIPriors program committee